# The Unrealised Potential for Predicting Pregnancy Complications in Women with Gestational Diabetes: A Systematic Review and Critical Appraisal

**DOI:** 10.3390/ijerph17093048

**Published:** 2020-04-27

**Authors:** Shamil D. Cooray, Lihini A. Wijeyaratne, Georgia Soldatos, John Allotey, Jacqueline A. Boyle, Helena J. Teede

**Affiliations:** 1Monash Centre for Health Research and Implementation, School of Public Health and Preventive Medicine, Monash University, Melbourne, VIC 3800, Australia; lw9767.2009@my.bristol.ac.uk (L.A.W.); Georgia.Soldatos@monash.edu (G.S.); Jacqueline.Boyle@monash.edu (J.A.B.); 2Diabetes Unit, Monash Health, Clayton, VIC 3168, Australia; 3Barts Research Centre for Women’s Health, Barts and the London School of Medicine and Dentistry, Queen Mary University of London, London E1 2AB, UK; j.allotey@qmul.ac.uk; 4Multidisciplinary Evidence Synthesis Hub, Queen Mary University of London, London E1 2AB, UK; 5Monash Women’s Program, Monash Health, Clayton, VIC 3168, Australia

**Keywords:** gestational diabetes, prediction, risk, prognosis, pregnancy complications, large-for-gestational age, neonatal hypoglycaemia, pre-eclampsia, shoulder dystocia, systematic review

## Abstract

Gestational diabetes (GDM) increases the risk of pregnancy complications. However, these risks are not the same for all affected women and may be mediated by inter-related factors including ethnicity, body mass index and gestational weight gain. This study was conducted to identify, compare, and critically appraise prognostic prediction models for pregnancy complications in women with gestational diabetes (GDM). A systematic review of prognostic prediction models for pregnancy complications in women with GDM was conducted. Critical appraisal was conducted using the prediction model risk of bias assessment tool (PROBAST). Five prediction modelling studies were identified, from which ten prognostic models primarily intended to predict pregnancy complications related to GDM were developed. While the composition of the pregnancy complications predicted varied, the delivery of a large-for-gestational age neonate was the subject of prediction in four studies, either alone or as a component of a composite outcome. Glycaemic measures and body mass index were selected as predictors in four studies. Model evaluation was limited to internal validation in four studies and not reported in the fifth. Performance was inadequately reported with no useful measures of calibration nor formal evaluation of clinical usefulness. Critical appraisal using PROBAST revealed that all studies were subject to a high risk of bias overall driven by methodologic limitations in statistical analysis. This review demonstrates the potential for prediction models to provide an individualised absolute risk of pregnancy complications for women affected by GDM. However, at present, a lack of external validation and high risk of bias limit clinical application. Future model development and validation should utilise the latest methodological advances in prediction modelling to achieve the evolution required to create a useful clinical tool. Such a tool may enhance clinical decision-making and support a risk-stratified approach to the management of GDM. Systematic review registration: PROSPERO CRD42019115223.

## 1. Introduction

Gestational diabetes (GDM) affects 7–20% of pregnancies and confers an increased risk of pregnancy complications with health consequences for both mother and baby. These risks are related to elevated glucose in GDM, but the relationship is complex, and an individual’s risk is modified by interrelated factors, including maternal weight [1,2], gestational weight gain [3], and ethnicity [4]. Accumulating empirical data suggests this phenotypic heterogeneity may be explained by multiple physiologic defects, demonstrable on sophisticated laboratory insulin secretion and sensitivity testing [5,6]. As a result of this heterogeneity, there is a continuum and breadth in the risk of pregnancy complications associated with contemporary definitions for this condition [7,8]. Therefore, for GDM, like in much of healthcare, there is a need to move from the current one-size-fits-all approach towards a personalised and risk-stratified model-of-care.

A personalised approach would stratify women with GDM by the estimated risk of pregnancy complications. Those at high risk would maximally benefit from the targeted delivery of evidence-based preventative and therapeutic interventions. Those at low risk would be spared unnecessary treatment and may be offered less intensive intervention. Accurate risk prediction models working within existing diagnostic definitions and utilising predictors readily available in routine care, could be implementable in clinical care and would be feasible and scalable. From a public health perspective, this could enable a risk-stratified approach and development of new models of care to better allocate scarce healthcare resources, imperative in the context of the increasing GDM prevalence [9,10,11,12].

Stratifying affected women by their risk of pregnancy complications requires a method to estimate the absolute risk of future events in an individual based on readily available characteristics, or a prediction model. Many fields of medicine have seen rapid growth in the development of prediction models. However, such models are rarely translated to clinical practice [13,14], and hence rarely positively influence patient care as their creators intended. A systematic review was conducted to establish the existing literature and inform progress towards optimal primary research in prediction modelling [15] and its translation into clinical care.

The aims of this systematic review were to: identify prognostic prediction models for pregnancy complications in women with GDM; describe characteristics of the identified prognostic prediction models qualitatively; compare the performance of identified prognostic prediction models quantitatively, with meta-analysis if appropriate, and critically assess the conduct and reporting of prediction modelling development methods.

## 2. Materials and Methods

A detailed description of the methods is available in the published protocol [16]. In summary, a systematic review of prediction modelling studies for pregnancy complications in women with GDM was conducted to identify all prediction models relevant to developing a risk-stratified approach to GDM. A sensitive search strategy was developed combining the Ingui filter for prediction modelling studies [17] as updated by Geersing [18] with keywords and subject headings for gestational diabetes and relevant pregnancy complications (Appendix A). A search of MEDLINE and Embase from inception to 16 August 2019 was executed. No limits on publication date nor language were applied. Study selection, data extraction and critical appraisal were conducted independently by two reviewers. Data extraction was conducted with guidance from the CHARMS checklist (checklist for critical appraisal and data extraction for systematic reviews of prediction modelling studies) [19]. Critical appraisal focusing on the risk of bias and concerns regarding applicability was conducted using the Prediction model Risk Of Bias overdiagnosis Tool (PROBAST) [20]. The systematic review protocol was registered with the International Prospective Register of Systematic Reviews (PROSPERO), number CRD42019115223.

## 3. Results

### 3.1. Study Selection

The search returned 12,161 unique records. Following title and abstract screening, the full text of 63 articles were assessed. Five studies meeting the selection criteria were included in this review (Figure 1) [21,22,23,24,25].

### 3.2. Study Characteristics

The five included studies reported the development of ten prediction models for pregnancy complications in women with GDM (Table 1). No validation studies were identified. The composition of the pregnancy complications predicted varied across the studies. Four studies reported the development of a single prediction model [22,23,24,25]. One study reported the development of six models, one for each of the six outcomes (primary caesarean delivery, birth injury, large-for-gestational-age (LGA), adiposity, hyperinsulinaemia, hypoglycaemia) [21]. In this review, these six models are presented collectively due to the shared model development process and methodological characteristics.

### 3.3. Characteristics of the Models

#### 3.3.1. Source of Data and Participants

Four prediction modelling studies had a retrospective study design using routinely collected data, three from a single centre [22,23,25] and one from multiple centres [24] (Table 2). One prediction modelling study used a historical cohort from a single centre of a multi-centre prospective observational study [21].

The study populations varied across studies (Table 3). Two studies included only women with GDM [23,24] while three also included women without GDM [21,22,25]. Diagnostic criteria for GDM varied by region. Uniquely, one study was a post hoc analysis of a historical cohort where participants and clinicians were blinded to oral glucose tolerance test (OGTT) results [21]. Hence, in this study, 10.5% of participants who would meet the International Association of Diabetes in Pregnancy Study Groups diagnostic criteria for GDM, did not receive any specific treatment for this condition. In the other studies, treatment for GDM followed a standardised institutional protocol [23,24,25] or was not reported [22].

Exclusion criteria were comparable across studies with the exception of the prospective observational study with blinded OGTT results [21] (Table 3). This study had more extensive exclusion criteria to reduce the likelihood of non-adherence to the study protocol. This prediction modelling study also excluded participants with missing data for predictors and those with non-Caucasian ethnicity.

#### 3.3.2. Outcome(s) to be Predicted

The pregnancy complications for prediction in women with GDM varied across studies (Table 4 and Figure 2).

The most common outcome to be predicted was the delivery of an LGA neonate, as it was included as an outcome in four of the five prediction modelling studies (Table 4 and Figure 2). In these four studies, LGA was defined as greater than the 90th percentile by gestational age. However, there was variation in the methods of standardization, with one study adjusting for maternal height and weight, ethnicity and parity [25] and the others adjusting for fetal sex and parity [24], fetal sex and ethnicity [22] or fetal sex alone [21]. Two of these studies included LGA neonates within a composite [22,24], it was handled as a single outcome in another study [21], and was a sole outcome for another [25].

Neonatal hypoglycaemia was selected as an outcome in three studies. It was defined as blood glucose < 2.2 mmol/L universally, however, there were variations in the timing of measures and mode of measurement were not defined in two studies [21,24]. Neonatal hyperinsulinaemia was selected as an outcome in two studies, defined as an elevated cord c-peptide level in one study [21] and as an elevated neonatal serum insulin measure in the other [24].

Two models predicted a composite outcome [22,24]. The first predicted a composite for adverse outcomes affecting both the mother and neonate [22]. The prediction of the second was limited to complications affecting the neonate [24]. However, this composite was more extensive, including 11 outcomes. Similar pregnancy complications were the subject of a third prediction modelling study, however here the outcomes were predicted in six discrete models for a single outcome rather than as a composite [21].

#### 3.3.3. Candidate Predictors

The number of candidate predictors investigated ranged from seven to 24 (Table 4). A variety of candidate predictors were chosen for model development with a tendency towards those that are routinely available in clinical practice via patient history or physical examination, or glycaemic measures available routinely via diagnostic testing for GDM. One study considered predictors from investigations that may not be available in routine care in all settings, serum analytes (first-trimester pregnancy-associated plasma protein A and second-trimester total human chorionic gonadotropin and inhibin-A serum) and fetal abdominal circumference from an obstetric ultrasound performed between 24 and 30 weeks gestation [25].

#### 3.3.4. Model Development

The presence and handling of missing data were not adequately reported in the four studies utilising routinely collected data [22,23,24,25] (Appendix A).

Statistical power varied across the studies, from two to 106 events per predictor (Table 4 and Figure 3).

Multivariable logistic regression was the most commonly used modelling method, described in four studies [21,22,23,25]. Notably, a tree-based approach, namely the recursive partitioning and amalgamation (RECPAM) method, was used in one study (Appendix A) [24]. In this study a binary decision-tree that uses answers from a series of yes/no questions about clinical characteristics was developed to predict an individual’s likely outcome. Continuous predictors were dichotomised using cut-points fitted to the development data in two studies [24,25]. Methods for the selection of predictors for inclusion in and during multivariable modelling varied across the studies (Appendix A).

#### 3.3.5. Predictors Selected in the Final Models

Ten predictors were selected for the final models across the five studies (Table 5 and Figure 4). Some measure of glycaemia and BMI was included in the final models of four of the five studies.

#### 3.3.6. Model Evaluation

Model performance was evaluated using the same dataset used to develop the model (apparent validation) in all but one study [25] (Table 5). In this study, results of internal validation based on resampling of the development dataset using bootstrapping was reported. No studies reported results of external validation as a measure of transportability of the developed model to new populations.

#### 3.3.7. Model Presentation

Three studies [23,24,25] included an alternative presentation format of the final models designed for ease of use and clinical application (Table 5).

### 3.4. Comparison of Predictive Performance

Model performance was most commonly reported in terms of discrimination with four of the studies reporting a concordance statistic (*c*-statistic) for their final models [21,22,23,25]. For these four models predicting a binary outcome, the *c*-statistic is equal to the area under the receiver operating characteristic (ROC) curve (AUC) and ranged from 0.517 to 0.911 (Table 5). Two studies reported that calibration was adequate, presenting non-significant findings for the Hosmer–Lemeshow goodness-of-fit test [22,23]. A meta-analysis of performance measures was not appropriate because the model development studies were not sufficiently homogenous with regard to the outcome to be predicted and no validation studies for a common prediction model were identified.

Four studies reported classification measures: sensitivity and specificity and/or positive and negative predictive values [21,22,23,25]. Cut-points were not determined a priori. In two studies they were determined by selecting a point on the ROC curve closest to the upper left corner, which maximises the Youden index [21,22]. The method for determination was not specified in the other two studies [23,25].

### 3.5. Risk of Bias and Concerns Regarding the Applicability of Models

As assessed using the PROBAST tool [20], all models had a high risk of bias driven by the analysis domain (Figure 5 and Appendix A). There was a high concern regarding the applicability of two studies to the systematic review question [21,22], as only a minority of the participants were affected by GDM. Moreover, in one of these studies [21], the exclusion of non-Caucasian women and women with a history of gestational diabetes requiring pharmacologic treatment may further limit its applicability.

## 4. Discussion

This systematic review identified five prediction modelling studies for pregnancy complications in women with GDM. Approaches to prediction varied, but the birth of an LGA neonate was the leading outcome, whether as part of a composite or singularly. Models seeking to predict a single outcome were more discriminatory than those predicting a composite outcome. Three predictors emerged in most models: glycaemic measures, BMI, and maternal age. Predictive performance was generally inadequately reported, and external validation was lacking. All models had a high risk of bias due to methodologic limitations in analysis as assessed by PROBAST.

### 4.1. Models Identified

Ten prediction models were developed by five prediction modelling studies, reflecting five distinct approaches to the clinical problem of quantifying the absolute risk of pregnancy complications in women with GDM. The literature is relatively lacking, compared to the related, but distinct literature on diagnostic prediction models for the development of GDM, with 17 models identified in a recent review [31]. However, interest in prognostic-based approaches to pregnancy risks in GDM is growing, with the first model published ten years ago and the later four within the last three years.

The model developed using a prospective cohort utilised an unselected population of pregnant women, of which 10.5% would meet current diagnostic criteria for GDM [21]. We acknowledge that the population in this prediction modelling study may not strictly meet the population criteria for eligibility of this review. However, we believe that the omission of this study would limit the value of this review, given its robust prospective study design and unique treatment-naïve study population. Furthermore, recognising that there is a continuum of risk for pregnancy complications related to GDM, this study is valuable because it facilitates feasibility assessment for a prediction model for pregnancy complications in women with hyperglycaemia independent of the consensus-based International Association of Diabetes and Pregnancy Study Group diagnostic threshold for GDM.

A recent prediction modelling study conducted by Barnes and colleagues [32] featured prominently at the title and abstract screening stage as it seemed to be especially applicable to the review question. This study developed and externally validated a model to predict the need for insulin therapy in women with GDM. Following model development, a post-hoc analysis found that the outcome of this model, the need for insulin therapy, was strongly associated with pregnancy complications. As pregnancy complication, the outcome of interest for this review was not the subject of this prediction modelling study it was ultimately excluded. We, however, note the close relationship of these outcomes and the relevance of this existing prediction modelling study to the overarching aim of developing a stratified model-of-care for women with GDM.

### 4.2. Characteristics of the Models

#### 4.2.1. Outcome(s) to Be Predicted

The delivery of an LGA neonate was the most common outcome predicted, offering three key advantages. Firstly, it reflects the classical maternal hyperglycaemia-fetal hyperinsulinaemia hypothesis, linking maternal hyperglycaemia to the LGA neonate, via transplacental glucose transport causing secondary fetal hyperinsulinaemia [33]. Secondly, although potentially too simplistic, excessive fetal growth is the unifying feature linking GDM to downstream pregnancy complications, such as failure to progress in labour, obstetric intervention, and shoulder dystocia. Thirdly, an LGA neonate with excess neonatal adiposity has poorer long-term metabolic health [34,35,36,37,38], with potential inter-generational implications [39].

Where multiple outcomes are potentially relevant, the prediction of a composite outcome, as in two of the models [22,24], may more accurately quantify multiple risks that concern women and clinicians and may be more translatable into clinical practice than a model predicting a single outcome. However, a poorly constructed composite outcome may be confusing and limit clinical application. Future model development should provide a clear rationale for the use and formulation of composite outcomes. There may be utility in heeding recommendations that the components of composite outcomes: (1) are of similar importance, (2) occur with similar frequency, and (3) are likely to have similar relative risk reductions (or predictive effects moving in the same direction) with similar underlying biology [40].

#### 4.2.2. Model Development

Two studies were inadequately powered with less than 10 events per predictor (EPP) [23,25] (Table 4 and Figure 3), generating significant risks for overfitting and consequently biased predictions [19]. The two studies which predicted composite outcome [22,24] were adequately powered for model development with more than twenty EPP [19]. This is an advantage of composite outcomes where event rates are low (Figure 4). The EPP for the six models developed in one study could not be calculated as the candidate predictors were not reported [21]. Although an EPP above 10–20 is traditionally advocated as the minimum sample size for model development [19], future studies should consider the recent proposal that a tailored sample size estimate may be advantageous in certain circumstances [41].

The dichotomisation of continuous predictor variables leads to substantial loss of information and is widely discouraged [42,43,44,45]. This is an inherent disadvantage of tree-based models included in this review [24] and was also notable in the model developed using classical regression methods [25]. In both models, continuous predictor variables were dichotomised using data-driven cut points, leading to high risks of bias [46]. In two models, continuous variables were dichotomised using cut-points which were pre-defined and independent to the development data [21,23], which leads to the loss of information but minimises the risk of bias. Continuous predictors were only handled optimally in one study [22]. Future model development studies should avoid dichotomising continuous predictors.

Selection of predictors is a key component of the model building at two stages, 1) inclusion in modelling and 2) during multivariable modelling. Where selection is based on associations with the outcome in the development dataset, there is a high risk that the developed model will be overfitted to this dataset. This was observed in three studies [22,23,25] and was inadequately reported in the other two [21,24]. In future studies, model generalisability may be improved by a priori selection of predictors for inclusion or selection independent of the predictor–outcome association, such as those based on clinical expertise [47].

#### 4.2.3. Predictors Selected in the Final Model

The predictors most commonly selected in final models, glycaemic measures (*n* = 4), BMI (*n* = 4), and age (*n* = 2) are routinely available in clinical practice. These predictors should be included in the set of candidate predictors evaluated in future model development studies.

#### 4.2.4. Model Evaluation

Model evaluation addresses two questions pivotal for the application of prediction models into clinical practice: (1) how accurate are its predictions and (2) how generalisable is it [48]. Accuracy relates to a model’s internal validity or “reproducibility” [45]. Internal validation techniques include apparent or bootstrap validation. Generalisability considers how well the model is likely to perform in a new but related population. It relates to a model’s external validity or transportability, tested by evaluating the model in a new population (external validation) [45]. External validation measures predictive performance and corrects internal validation for the inherent optimism of a model being overfitted to the development dataset. As such, it provides a more realistic measure of predictive performance and quantifies the “transportability” of a model to other populations. No studies identified here reported external validation.

Three studies reported measures of model performance using apparent validation [21,22,23]. Here model performance is evaluated against the sample from which it was developed. The utility is limited, as performance is biased towards overestimating model performance in new populations. This risk of bias is further exaggerated with the small development datasets noted here.

One reviewed study assessed internal validity using simple bootstrapping validation, with the model repeatedly fitted to 1000 bootstrap samples and the average area under the ROC calculated as an estimate of future performance of the model in other populations [25]. This technique for internal validation is preferable to apparent validation conducted in the other studies, with less biased performance estimates. It is also preferable to split-sample validation, as data available for development and validation are maximised. Ultimately, however, external validation is essential to establish the confidence in a model required for clinical application.

The included studies did not report any formal evaluation of the clinical usefulness of the developed models. This could include the net-benefit realised by using the model’s predictions to guide clinical decision-making [49]. Future studies may consider decision curve analysis [50] to quantify the clinical value of developed models as is increasingly reported in the literature [51] and recommended [52,53,54].

#### 4.2.5. Model Presentation

A clinical prediction model is developed with the overarching premise to be applied to clinical care to improve outcomes. Hence, developed models should be presented in a format fit for this purpose [55]. Two studies presented simplified scoring systems [23,25], and one presented a simple decision tree [24], all readily applicable to clinical care. The other studies reported regression coefficients for included predictors but did not present baseline components or regression equations [21,22], and as such do not facilitate the clinical application of the models reported.

### 4.3. Comparison of Predictive Performance

The overall performance of prediction models is traditionally quantified with two essential measures of predictive performance: discrimination and calibration [49]. Consistent with general prediction model literature [48], performance was incompletely evaluated in prediction models identified in this review (Table 5). Where reported, discrimination was evaluated using the *c*-statistic and was graphically presented with ROC curves. Calibration was evaluated using the Hosmer–Lemeshow goodness of fit test.

Discrimination was highest for models predicting single outcome versus composite outcomes overall as observed in the risk score for pre-eclampsia [23] and the fetal overgrowth index [25]. Models with single outcomes also had minimal loss in performance when adapted into simplified clinical tools. Discrimination was more limited with composite outcomes [22] and was not reported in the model developed using a decision-tree for a composite of neonatal complications [24]. In the fifth study, discrimination varied considerably for the six independent models for single pregnancy complications [21].

Calibration was evaluated in two studies using the Hosmer–Lemeshow goodness-of-fit test, suggesting an agreement between predicted and observed probabilities [22,23]. However, this method alone does not capture the magnitude or direction of miscalibration, limiting clinical utility. For instance, a model with accurate predictions in the intermediate risk range, can consistently overestimate in the low-risk range and underestimate in the high-risk range, limiting clinical usefulness. Another limitation of the Hosmer–Lemeshow test is that it is strongly related to sample size and is usually non-significant for small data sets, and conversely significant for large data sets [46]. Future work should ideally report calibration graphically (calibration plot) or in a tabular format, comparing predicted probabilities to observed outcome frequencies, and allowing the model’s performance to be assessed at clinically relevant risk ranges [46,56].

### 4.4. Risk of Bias and Concerns Regarding the Applicability of Models

#### 4.4.1. Risk of Bias

Collectively, the high risk of bias for prognostic prediction models for women with GDM limits generalisability (Figure 5 and Appendix A), driven by analysis methods and overfitting of models to development datasets. This is reflective of the rapid evolution of prediction modelling methodology. The risk of bias of future prediction models may be reduced by addressing the findings of this review and referring to relevant guidelines such as the TRIPOD (transparent reporting of a multivariable prediction model for individual prognosis or diagnosis) statement [57].

#### 4.4.2. Concerns Regarding Applicability

The applicability of identified models varied based on the participants in the development dataset (Figure 5 and Appendix A). Two models sought to develop prediction models for pregnancy complications as a basis for defining alternative diagnostic criteria, including women both with and without GDM [21,22]. Their aims differed from this review, the prediction of pregnancy complications in women with GDM limiting applicability.

### 4.5. Strengths and Limitations of this Review

To our knowledge, this is the first systematic review of prediction models for pregnancy complications in GDM. Strengths of this review include rigorous methods and a sensitive search strategy utilising standardised and validated search terms across the entirety of the two leading databases of biomedical literature since their inception. Bias was minimised with prospective registration and peer-reviewed publication protocol [16]. The risk of bias and applicability of included models was systematically and objectively evaluated using PROBAST [20], a robust and for-purpose tool.

Limitations included the inability to synthesize the quantitative characteristics of included models due to the heterogeneity of included studies. However, the results of the systematic search support this broad approach. We note that, despite the clinical imperative, there are only five prediction modelling studies for pregnancy complications in GDM that met our eligibility criteria in the indexed medical literature to date.

Finally, given the heterogeneity of this condition and the variety of diagnostic approaches currently used, a developed model cannot be assumed to perform equally well (or poorly) in a new, but related population. Clinical prediction modelling is an iterative multi-stage process [58]. External validation in a range of new but related populations, both geographically and temporally, is crucial but, as this review suggests, frequently neglected. Such external validation may facilitate model updating which, by addressing the characteristics of the local population and clinical practice, is likely to optimise model performance [59].

## 5. Conclusions

This review demonstrates the potential for prediction models to provide an individualised absolute risk of pregnancy complications for women affected by GDM. However, limitations in current models have been identified and this emphasises that future model development and validation would benefit from the application of methodologic advances in this rapidly evolving field. External validation, including appropriate reporting of calibration and formal evaluation of clinical usefulness with decision curve-analysis, will significantly assist the translation of promising statistical models into a useful clinical tool. Such a tool would be capable of improving outcomes for women with GDM by enhancing clinical decision-making and facilitating the stratification of affected women by their risk of pregnancy complications, thus enabling a personalised model-of-care.

## Figures and Tables

**Figure 1 ijerph-17-03048-f001:**
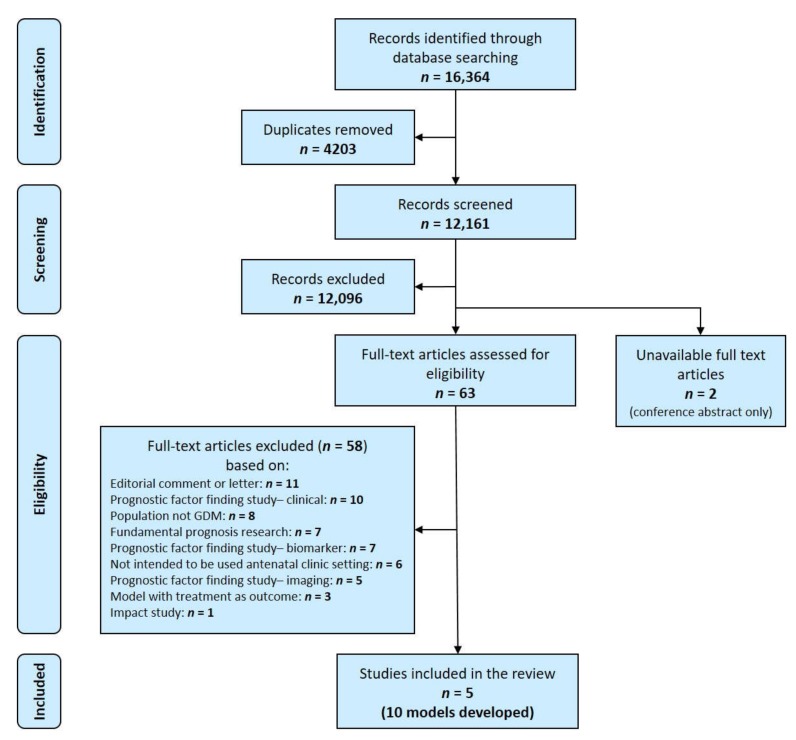
Flow diagram of the identification, screening and eligibility assessment of the literature for prediction models for pregnancy complications in women with gestational diabetes.

**Figure 2 ijerph-17-03048-f002:**
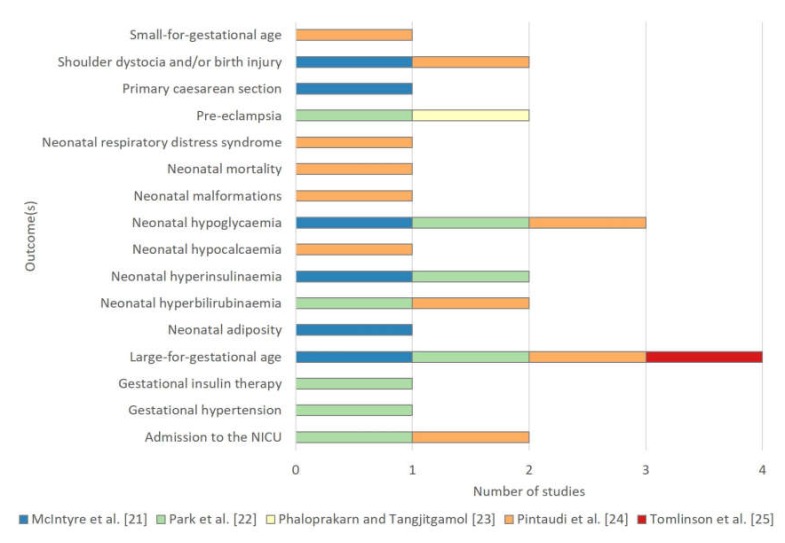
Outcome(s) to be predicted by studies for pregnancy complications in women with gestational diabetes.

**Figure 3 ijerph-17-03048-f003:**
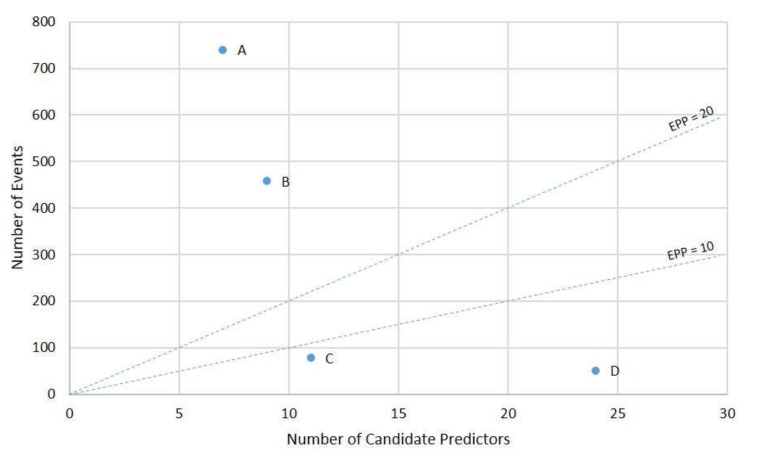
Events per predictor for prediction modelling studies for pregnancy complications in women with GDM. The events per predictor (EPP) are shown, where A indicates the EPP for Pintaudi et al. [24]; B, Park et al. [22]; C, Phaloprakarn and Tangjitgamol [23]; D, Tomlinson et al. [25]. The EPP could not be calculated for McIntyre et al. [21] An EPP above 10 to 20 is regarded as the minimum sample size for model development [19]. This graphical presentation format was adapted from Ensor and colleagues [30].

**Figure 4 ijerph-17-03048-f004:**
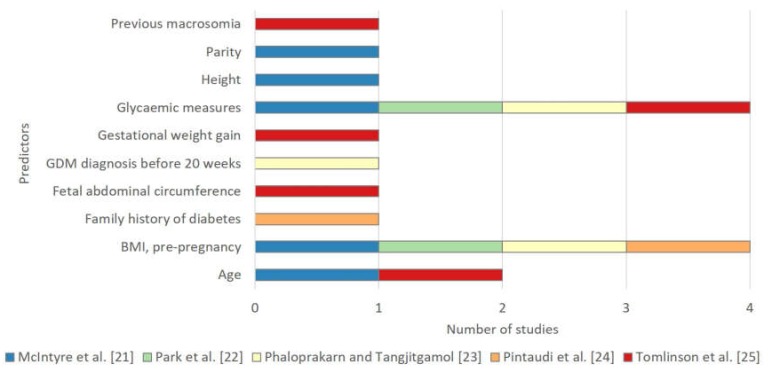
Selected predictors in final models for pregnancy complications in women with gestational diabetes.

**Figure 5 ijerph-17-03048-f005:**
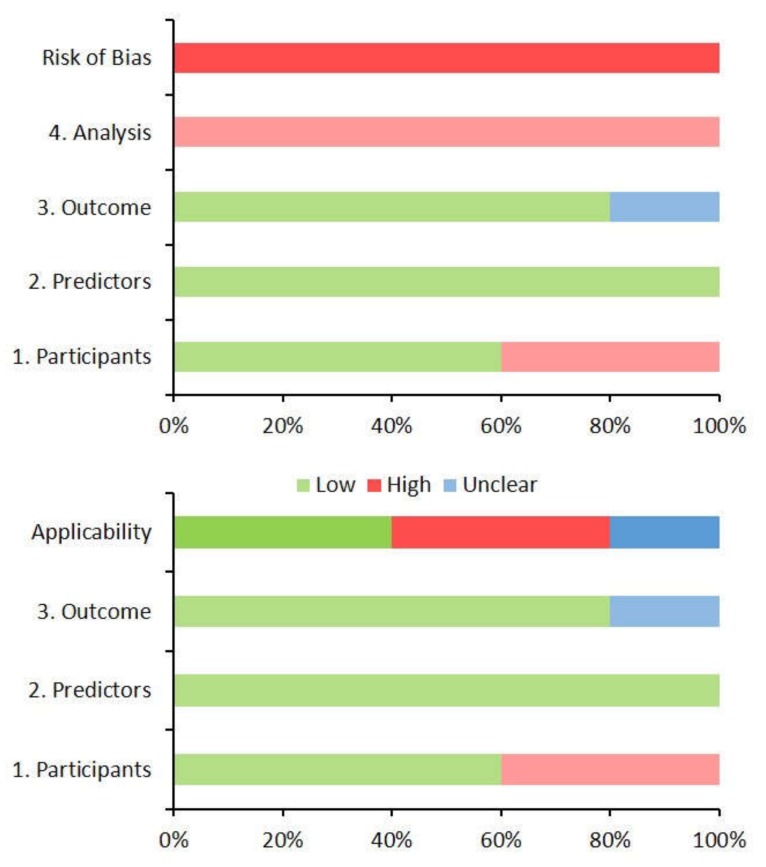
The risk of bias and concern regarding the applicability of the models developed in the five prediction modelling studies for pregnancy complications in women with gestational diabetes using the Prediction model Risk of Bias Assessment Tool (PROBAST). The *x*-axes display the proportion of studies rated by level of concern (low, high or unclear) for risk of bias or applicability for each domain.

**Table 1 ijerph-17-03048-t001:** The ten models developed by the five prediction modelling studies for pregnancy complications in women with gestational diabetes.

Study	Type of Prediction Modelling Study	Model(s)
McIntyre et al. [21]	Development	A. A risk engine relating maternal glycaemia and body mass index to pregnancy outcomes with a model developed for each of: Primary caesarean delivery Birth injury Large-for-gestational age Adiposity Hyperinsulinaemia Hypoglycaemia
Park et al. [22]	Development	B. Screening tool for predicting adverse outcomes ^a^ of GDM
Phaloprakarn and Tangjitgamol [23]	Development	C. A risk score based on clinical characteristics of GDM women for the development of preeclampsia
Pintaudi et al. [24]	Development	D. Subgroups at different risks of developing the composite adverse neonatal outcome ^b^ by RECPAM analysis
Tomlinson et al. [25]	Development	E. The fetal overgrowth index

Abbreviations: GDM, gestational diabetes; RECPAM, RECursive Partitioning and Amalgamation; ^a^ Adverse outcomes were neonatal hypoglycaemia, hyperbilirubinemia, and hyperinsulinemia; admission to the neonatal intensive care unit; large-for-gestational age; gestational insulin therapy; preeclampsia; and gestational hypertension; ^b^ Neonatal adverse outcomes were fetal growth large or small for gestational age, mortality (neonatal deaths and stillbirths), malformations, shoulder dystocia, neonatal intensive care unit need, hypoglycaemia, hypocalcaemia, hyperbilirubinemia, and respiratory distress syndrome.

**Table 2 ijerph-17-03048-t002:** Source of data and characteristics of studies used to develop models for pregnancy complications in women with gestational diabetes.

Model	Source of Data		Study Setting	Study Dates	Sample Size
A risk engine relating maternal glycaemia and body mass index to pregnancy outcomes [21]	Prospective cohort (post hoc analysis)		Single centre The antenatal clinic, Mater Misericordiae Mothers’ Hospital, Australia	Jul 2000– Apr 2006	*n* = 1248
Screening tool for predicting adverse outcomes of GDM [22]	Retrospective cohort		Single centre Admitted patients, Severance Hospital, South Korea	Mar 2001–Apr 2013	*n* = 802 (306 in GDM group)
A risk score based on clinical characteristics of GDM women for the development of preeclampsia [23]	Retrospective cohort		Single centre The antenatal clinic, Vajira Hospital, Thailand	Jan 2003– Feb 2008	*n* = 813
Subgroups at different risks of developing the composite adverse neonatal outcome [24]	Retrospective cohort		Multi-centre Specialised GDM clinics, Various hospitals, Italy	Jan 2012–May 2015	*n* = 2736
The fetal overgrowth index [25]	Retrospective cohort		Single centre Collaborative diabetes in pregnancy program, South Shore Hospital, USA	Mar 2010–May 2012	*n* = 275

Abbreviation: GDM, gestational diabetes.

**Table 3 ijerph-17-03048-t003:** Clinical characteristics of the study populations used to develop models for pregnancy complications in women with gestational diabetes.

Models	Inclusion Criteria	Exclusion Criteria	Nulliparous	Ethnicity	Diagnostic Criteria Used For GDM	Treatment Status
A risk engine relating maternal glycaemia and body mass index to pregnancy outcomes [21]	Pregnant women enrolled in HAPO study ^a^	Multiple pregnancy, stillbirth, congenital anomaly, non-Caucasian mother, birth < 33 weeks gestation, missing data for key independent variables, age ≤ 18 years, uncertainty of gestational age, inability to complete oral glucose-tolerance test within 32 weeks gestation, assisted conception, previous diabetes requiring pharmacologic treatment or infection with hepatitis B or C virus	54.5%	100% Caucasian	NA (Participants blinded to results of OGTT)	No GDM treatment (patients and clinicians blinded to OGTT result)
Screening tool for predicting adverse outcomes of GDM [22]	Two groups: 1) Women with GDM, 2) pregnant women with false-positive glucose challenge tests	Multiple pregnancy, pre-gestational diabetes, diagnosis with GDM at <24 weeks gestation, anomalous foetuses, chronic hypertension	NR	NR	Universal screening with ACOG approach ^b^	Treatment not reported
A risk score based on clinical characteristics of GDM women for the development of preeclampsia [23]	Women with GDM	Multiple pregnancy, risk factors for pre-eclampsia, smoking	40.0%	92% Thai, 8% South East Asian	Universal screening with ACOG approach ^b^	GDM treatment as per standardised institutional protocol
Subgroups at different risks of developing the composite adverse neonatal outcome [24]	Women with GDM	Multiple pregnancy	45.3%	44.8% Caucasian	Risk factor-based screening with IADPSG approach ^c^	GDM treatment as per standardised institutional protocol
The fetal overgrowth index [25]	Women with pre-gestational diabetes or GDM	Multiple pregnancy, birth <20 weeks gestation	50.5%	82% White	Universal screening with ACOG approach ^b^	GDM treatment as per standardised institutional protocol

Abbreviations: GDM, gestational diabetes; HAPO, Hyperglycaemic and adverse pregnancy outcomes; IADPSG, International Association of Diabetes in Pregnancy Study Groups; NA, not applicable; OGTT, oral glucose tolerance test; NR, not reported; ACOG, American College of Obstetricians and Gynaecologists; ^a^ The HAPO study [26] included all pregnant women unless they had one or more exclusion criteria listed above; ^b^ ACOG approach = two-step procedure using a screening 50 g glucose challenge test (GCT) with abnormal ≥ 140 mg/dL (7.8 mmol/L) and diagnostic 100 g 3 h oral glucose tolerance test (OGTT) with two or more values above the Carpenter-Coustan cut-offs [27] considered abnormal (≥ 95 mg/dL [5.3 mmol/L] at baseline, ≥ 180 mg/dL [10.0 mmol/L] after 1 h post-load, ≥ 155 mg/dL [8.6 mmol/L] after 2 h post-load, ≥ 140 mg/dL [7.8 mmol/L] after 3 h post-load) [28]; ^c^ IADPSG approach = risk factor-based screening with a 75 g 2 h OGTT with one or more values above the IADPSG cut-offs considered abnormal (≥ 92 mg/dL [5.1 mmol/L] at baseline, ≥ 180 mg/dL [10.0 mmol/L] at 1 h post-load, ≥ 153 mg/dL [8.5 mmol/L] at 2 h post-load) [29].

**Table 4 ijerph-17-03048-t004:** Outcome(s) to be predicted and candidate predictors in models for pregnancy complications in women with gestational diabetes.

Model	Outcome(s) to Be Predicted			Candidate Predictors		Events per Predictor
	Type	Event	Number of Events	Number	Type	
A risk engine relating maternal glycaemia and body mass index to pregnancy outcomes [21]	Single	(1) primary caesarean delivery	241	NR	NR	─
		(2) birth injury (including shoulder dystocia)	29			─
		(3) LGA	175			─
		(4) neonatal adiposity	100			─
		(5) neonatal hyperinsulinemia	76			─
		(6) neonatal hypoglycaemia	73			─
Screening tool for predicting adverse outcomes of GDM [22]	Composite	“adverse outcomes of GDM”: neonatal hypoglycaemia, hyperbilirubinemia, and hyperinsulinemia; admission to the NICU; LGA; gestational insulin therapy; preeclampsia; gestational hypertension	458	9	demographics, patient history, physical examination, investigations	51 ^a^
A risk score based on clinical characteristics of GDM women for the development of preeclampsia [23]	Single	preeclampsia	78	11	demographics, patient history, physical examination, investigations, disease characteristics	7
Subgroups at different risks of developing the composite adverse neonatal outcome [24]	Composite	“neonatal adverse outcome”: fetal growth large or small for gestational age, mortality (neonatal deaths and stillbirths), malformations, shoulder dystocia, NICU need, hypoglycaemia, hypocalcaemia, hyperbilirubinemia, and respiratory distress syndrome	740	7	demographics, patient history, physical examination, investigations	106
The fetal overgrowth index [25]	Single	fetal overgrowth: birthweight ≥ 90th gestational-related optimal weight centile	51	24	demographics, patient history, physical examination, investigations	2

Abbreviations: GDM, gestational diabetes; LGA, large-for-gestational age; NICU, neonatal intensive care unit; NR = not reported; ^a^ Calculated for the entire study population rather than the GDM only group.

**Table 5 ijerph-17-03048-t005:** The selected predictors, presentation format and performance of models for pregnancy complications in women with gestational diabetes.

Model	Selected Predictors	Presentation Format	Evaluation	Performance	
				Calibration	Discrimination (*c*-statistic ^a^ [95% CI])
A risk engine relating maternal glycaemia and body mass index to pregnancy outcomes ^b^ [21]	Fasting, one hour, two hour OGTT results ^c^, age, height, BMI at time of OGTT, parity	Regression coefficients without baseline components	Internal validation (apparent)	NR	Primary caesarean delivery: 0.694 (0.661–0.727) Birth injury: 0.699 (0.568–0.830) LGA: 0.654 (0.624–0.684) Adiposity 0.608 (0.570–0.645) Hyperinsulinaemia 0.601 (0.545–0.657) Hypoglycaemia 0.574 (0.517–0.632)
Screening tool for predicting adverse outcomes of GDM [22]	BMI at time of diagnosis, fasting glucose from OGTT ^d^	Regression coefficients without baseline components	Internal validation (apparent)	GOF (*p* = 0.27)	0.642 (NR)
A risk score based on clinical characteristics of GDM women for the development of preeclampsia [23]	First trimester BMI ≥ 27 kg/m^2^, GDM diagnosed within 20 weeks of gestation, poor glycaemic control ^e^	Simplified scoring system	Internal validation (apparent)	GOF (*p* = 0.792)	0.911 (0.877–0.946)
Subgroups at different risks of developing the composite adverse neonatal outcome [24]	Pre-pregnancy BMI, family history of diabetes	Decision tree consisting of four patient subgroup classes	NR	NR	NR
The fetal overgrowth index [25]	High fasting glucose ^f^, enlarged abdominal circumference ^g^, excessive weight gain ^h^, history of macrosomia ^i^, Age ≤ 30	Simplified scoring system	Internal validation (bootstrapping)	NR	0.89 (0.888–0.891)

Abbreviations: *c*-statistic, concordance statistic; CI, confidence interval, BMI, body mass index; GDM, gestational diabetes; GOF, goodness of fit; LGA, large-for-gestational-age; NR, not reported; OGTT, oral glucose tolerance test; ^a^ The concordance statistic is equal to the area under the receiver operating characteristic curve for models predicting binary outcomes; ^b^ This study presented results for each of eight fixed sets of predictors (designated models A to H) from the following: fasting, 1-hour post-load (1 h) and 2-hour post-load (2 h) OGTT results, haemoglobin A1c, age, height, BMI at time of OGTT (24–32 weeks gestation) and parity. Six of the eight sets of predictors (Models A to F) only included glycaemic measures as predictors—fasting, 1 h and 2 h glucose levels from an OGTT in varying combinations or averaged, or haemoglobin A1c. Two sets of predictors (Models G and H) combined four clinical characteristics (age, height, BMI and parity) with OGTT results, as individual components or averaged respectively. Of the eight sets of predictors evaluated the authors nominated model G (individual OGTT components with clinical characteristics) as having the best predictive ability and presented the models using this set of predictors most completely, reporting standardised coefficients for each. Hence, for comparison, we considered these models to represent the “final” model in their development process and hence, it is these results that are presented; ^c^ Results from a 75 g, 2-hour OGTT undertaken at 24 to 32 weeks gestation; ^d^ Results from a 100 g, 3-hour OGTT undertaken at 24 to 28 weeks gestation; ^e^ Poor glycaemic control defined as ≥ 2 separate occasions of fasting glucose ≥ 5.8 mmol/L and/or 2 postprandial glucose ≥ 6.7 mmol/L after GDM treatment; ^f^ High fasting glucose defined as fasting glucose at 24 to 30 weeks (either serum value derived from OGTT or mean fasting capillary blood glucose over 1 week) ≥ 5.6 mmol/L (100 mg/dL); ^g^ Enlarged abdominal circumference defined as ≥ 90th percentile on ultrasound between 24 and 30 weeks; ^h^ Excessive weight gain defined as weight gain in second and third trimester ≥ 0.3 lb/week above the Institute of Medicine (BMI-based) goal range; ^i^ History of macrosomia defined as prior infant birthweight > 4 kg.

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
