# Peer review of "The Unrealised Potential for Predicting Pregnancy Complications in Women with Gestational Diabetes: A Systematic Review and Critical Appraisal"

_ijerph, 2020, doi:10.3390/ijerph17093048_

Round 1

Reviewer 1 Report

Gestational diabetes mellitus (GDM) is associated with increased risk of maternal complications, adverse pregnancy and neonatal outcomes (macrosomy, preeclampsia, intrauterine growth restriction, neonatal hypoglycemia, hyperbilirubinemia, hypocalcemia, respiratory distress syndrome, polycythemia). Clinical prediction of these complications gives the possibility to perform preventative and therapeutic interventions. The most important problem is a prediction of large for gestational age newborns (LGA) as a cause of failure to progress in labour, obstetric intervention and shoulder dystocia.

In clinical practice the introduction of a simple predictive model with widely available clinical factors may be very useful to identify those pregnancies that are at increased risk of complications. The highest risk pregnancies may benefit from close monitoring, more stringent glycemic control, or other interventions while healthcare resources may be used more sparingly in those at lower risk.

In this study a systematic review of prognostic prediction models for pregnancy complications in GDM was performed. The five studies reported the development of ten prediction models for pregnancy complications in women with GDM were analyzed. The predictors most commonly used in final models are glycemic measures, BMI and age and these predictors should be used in future model development studies.

This review has demonstrated the feasibility of prediction modelling to provide an individualized absolute risk of pregnancy complications for women affected by GDM.

The most common outcome to be predicted was the delivery of an LGA neonate. It reflects the classical maternal hyperglycemia-fetal hyperinsulinemia hypothesis, linking maternal hyperglycemia to the LGA neonate, via transplacental glucose transport causing secondary fetal hyperinsulinemia.

This study is technically very well-performed and the conclusion is interesting and informative.

In my opinion some minor point should be additionally explained or changed:

In the limitations of this review it should be highlighted, that the risk of pregnancy complications for GDM women has to be individualized. There are no general-purpose criteria for all populations. In analyzed studies oral glucose tolerance test was performed in different periods of pregnancy (24-32 and 24-28 weeks of gestation), in different populations, and different tests for GDM were used. Diagnostic criteria for GDM varied by region.  We have to also remember about differences between populations in prevalence of obesity. And that's why it's difficult to find a universal model that can be used around the world. Rather, the regional models could be implemented.

Reviewer 2 Report

In this report Cooray et al. provide a systematic review and critical appraisal focused on gestational diabetes and its related complications. Advancing the knowledge of gestational diabetes is a topic of scientific interest. This reviewer finds surprising that authors submit a review when in depth in silico research has been performed. The article even has materials and methods, which refers to a recent article published by the authors on a similar topic. Authors finally focus their efforts in 5 articles that fulfilled the selection criteria. Authors describe prediction models and include several limitations of the analysis performed. This reviewer has no major criticisms to procedures described. However, given the numerous limitations, it is not clear for this reviewer how this work could improve the basic research, clinical research of clinical practice. This limits the enthusiasm of this reviewer this line of work.

Author Response

Dear Reviewer,

We have responded in detail to your comments. Please see the attachment.

Kind regards,

Shamil on behalf of Professor Teede and colleagues

Reviewer 3 Report

ijerph-757316

The unrealised potential for predicting pregnancy complications in women with gestational diabetes: A systematic review and critical appraisal

Summary.  The manuscript is an assessment of existing prognostic prediction models for pregnancy complications in women with gestational diabetes (GDM).  The authors correctly note that the prevalence of GDM is increasing, and thus the importance of being able to predict the probability of related pregnancy complications.  Despite a large body of relevant literature on the subject, the authors found only five published studies that met reasonable selection criteria.  The authors provide thorough details of a systematic assessment of the methodologies, potential biases, and applicability of each study.  The manuscript has only a small number of substantive concerns, as well as minor concerns related to editing and presentation.  Please note that the manuscript I received had inconsistent page and line numbering; I have tried as much as possible to clarify the specific areas being discussed below.

Concerns

  1. The authors reference four distinct aims of the review: identifying prediction models in the literature, describing characteristics of the models, comparing performance, and critically assessing their conduct and reporting [Introduction, page 2, last paragraph]. The critical assessment aim apparently relates to the Critical Appraisal mentioned in the study’s title, and also presumably the reference to “This systematic review demonstrates the feasibility of developing prediction models for pregnancy complications in women with GDM” [Discussion, page 2 of 27, line 147].   While the feasibility of developing prediction models may be an important question it is not resolved here.  If the key aspect of “feasibility” is the ability to create a valid and robust mathematical prediction model, the multiple flaws in the reviewed studies would seem to indicate a lack of feasibility.  A thorough (and explained) assessment of feasibility would seem to be a separate study.
  2. In Figure 1, the reasons for excluding 99.5% of screened records should be provided.
  3. Section 3.3.1 (Source of data and participants, lines 19-20) refers to “more extensive exclusion criteria in one study.” The effect of this on study bias could be developed further when discussing the PROBAST bias results.
  4. The Discussion section (particularly but not uniquely Section 4.2.1 Outcomes to be predicted, page 3 of 27, lines 183-200) contains details that would fit better into a Background section on the methodology and assessment of models. As presented, they detract from the overall discussion of the strengths and weaknesses of the five studies.
  5. Similarly, Section 4.2.3 (Predictors selected in the final model, lines 230-239) engages in an excessive rationale of the study results, above and beyond the purpose of a systematic review focused on methodology.

Minor concerns

  1. The abstract states “…from which ten prognostic models primarily intended to predict pregnancy complications related to GDM were developed.” Clarifying that these models involved differing and/or multiple pregnancy complications (outcomes) would help orient the reader; this is also relevant in section 3.2 (Study characteristics).
  2. Figure 1 notes that 2 full-text articles were excluded because they were not full text, which seems contradictory.
  3. Table 1 and elsewhere in the manuscript refers to “risk engine,” which is a term that may not be familiar to some readers, and merits a brief footnote of explanation.
  4. Table 1 also introduces the outcome measures for the reviewed studies. It would helpful to the reader to have the “adverse outcomes of GDM” outcome (Park et al.) and “composite adverse neonatal outcome” (Pintaudi et al.) defined here rather than waiting for Table 4, as they are referred to multiple times before then.
  5. Table 3 unnecessarily describes HAPO study exclusion criteria twice – in the table and in the footnotes. The footnotes only need references to criteria that were not mentioned in the table.
  6. Figures 2 and 4 identify the studies by author names, whereas the reader has been tracking them by citation number; it would be helpful to include both, as is done in Figure 3. Also, the two shades of blue are difficult to distinguish, and overall the use of color coding is problematic for readers with a black and white printed copy.
  7. Section 3.3.2 (Outcome(s) to be predicted, lines 41-45) includes a very long sentence. It might be more easily digestible if broken at the comma between “…by gestational age, however there was…”.  Also in this sentence, the first “or” is not necessary.
  8. Table S2 (referred to in Section 3.3.4, Model development) uses different citation numbers; cross-reading them would be helped by including author names (or the original citation numbers) in the table. It is not immediately obvious which of the two with details was considered the adequate one (line 73).
  9. The reference to the RECPAM method (Section 3.3.4, line 85) would benefit from a brief explanatory (informative) footnote.
  10. Continuing in this section (lines 86-87), the studies are identified by authors as well as citation numbers; this is not consistent with the rest of the document.
  11. Figure 4 has poor resolution.
  12. Table S3 appears to have errors in the Overall – ROB column; all of the studies are noted as having low or no concerns. This does not seem to follow from the previous ROB columns.
  13. The term -its’- is not an actual word (Section 4.2.4 Model evaluation, line 243).
  14. Section 4.5 (Strengths and limitations of this review) should be clarified – there were only five prediction modelling studies *that met your review criteria* (page 5 of 27, lines 323-324).

Author Response

(The authors gave the same response as above.)

Round 2

Reviewer 2 Report

This reviewer does not have any new criticism or comment.